# Methods for Fungicide Efficacy Screenings: Multiwell Testing Procedures for the Oomycetes *Phytophthora infestans* and *Pythium ultimum*

**DOI:** 10.3390/microorganisms11020350

**Published:** 2023-01-31

**Authors:** Demetrio Marcianò, Silvia Laura Toffolatti

**Affiliations:** Dipartimento di Scienze Agrarie ed Ambientali, Università degli Studi di Milano, 20133 Milan, Italy

**Keywords:** oomycetes, plant pathogens, fungicide efficacy, statistical analyses

## Abstract

Oomycetes-borne diseases represent a serious problem for agriculture sustainability due to the high use of chemical products employed for their control. In recent years, increasing concerns on side effects associated with fungicide utilization have led to the reduction of the permissible modes of action, with the remaining ones continuously threatened by the increase of resistant strains in the pathogen populations. In this context, it is mandatory to develop new generation fungicides characterized by high specificity towards the target species and low environmental impact to guarantee the sustainability, productivity, and quality of food production. Fungicide discovery is a lengthy and costly process, and despite these urgent needs, poor description and formalization of high-throughput methodologies for screening the efficacy of active compounds are commonly reported for these kinds of organisms. In this study, a comprehensive picture of two high-throughput practices for efficient fungicide screening against plant-pathogenic oomycetes has been provided. Different protocols using multiwell plates were validated on approved crop protection products using *Phytophthora infestans* and *Pythium ultimum* as the model species. In addition, detailed statistical inputs useful for the analysis of data related to the efficacy of screenings are included.

## 1. Introduction

Fungicides represent a key input in crop protection to ensure the productivity and quality of agricultural products. However, the progressive reduction in the number of active substances available [1], and the increase of resistant strains in pathogens’ field populations are challenging the exploitation of the approved substances [2]. Anthropogenic activities can heavily affect the agroecosystem by favoring the spread of plant pathogens into new areas and increasing the selection pressure for individuals with higher virulence or resistance to fungicides [3]. This is particularly true for oomycetes, fungal-like eukaryotes belonging to the TSAR supergroup [4], which include many species that represent a threat to forestry, agriculture, and aquaculture [1,5], causing annual losses in the order of tens of billions of dollars [6]. Moreover, the market value of active ingredients available for oomycete control accounted for hundreds of millions of dollars in the USA market only for 2011, thus suggesting the relevant economic impact associated with these organisms [7].

*Phytophthora infestans* Mont. (de Bary) represents one of the most destructive, and hence, studied species among plant-pathogenic oomycetes. This organism is responsible for tomato and potato late blight [8,9,10], with economic losses estimated at billions of dollars every year [11]. Infections associated with this hemibiotrophic pathogen are mostly imputable to asexual spores (zoospores produced in sporangia) and mycelium in infected tubers [8,11]. The disease control associated with late blight mainly relies on the use of fungicides [12]. Another important phytopathogenic oomycete is *Pythium ultimum* Trow var. *ultimum*, which is the most studied species of its genus. It is a ubiquitous necrotrophic pathogen, causing damping off and root rot on more than 300 different hosts, including staple crops such as corn and wheat [10]. In this case, infections are mainly carried out by sporangia-like hyphal swellings [13]. Chemical control, routinely practiced through soil treatments, is not always feasible due to economic and ecological reasons [14].

Due to the great economic importance of oomycetes, in recent years, several new active molecules against oomycetes have been proposed [15,16,17,18,19], and besides traditional research workflows, new biotechnology-based approaches to control oomycetes, such as small RNAs and short peptides, are attracting interest [20,21]. In this context, screening candidates and existing molecules for efficacy against oomycetes plays a key role in fungicide discovery and resistance monitoring. The development of fast, high-throughput, and unbiased protocols to determine the efficacy of anti-oomycete compounds would allow an increase in the number of molecules screened and, meanwhile, reduce the amount of substances needed, both limiting the costs and the operator’s exposure to potential chemical risks. The last issue is particularly relevant at the preliminary screening level of fungicide discovery, where the custom synthesized molecules are poorly characterized for human health safety due to the high costs for the synthesis [22].

The amended agar medium (AAM) represents the gold standard methodology employed to test oomycetes for their sensitivity to active compounds. In this kind of assay, the mycelial growth is measured by referring to the colony diameter on Petri plates, using mycelium plugs as the source of inoculum on culture media amended with inhibitory molecules [23]. Starting from early 2000, different protocols for the high-throughput evaluation of molecules with anti-oomycete activity, mainly designed for species belonging to *Phytophthora* and *Pythium* genera, have been described (Table 1). However, these studies were often meant to characterize the activity of different putative inhibitory molecules, and neither comparison with the gold standard (AAM) nor with a known effective fungicide was included in the assays.

For instance, only a few applied studies on *P. infestans* reported the use of high-throughput methodologies based on optical density (OD) measurements, using zoospores from axenic culture as the source of inoculum [25,28,34]. However, recent studies have pointed out that spores from axenic cultures can exhibit reduced rates of zoospore release and encystment, with a great variation associated with laboratory practices and spore handling [12,35]. For these reasons, the use of sporangia as the source of inoculum is advisable to increase the reproducibility and to avoid possible biases in the initial absorbance measures due to the presence of active swimming of zoospores [36]. 

Spores represent a desirable source of inoculum, since they can be easily quantified. However, *in vitro* production of such structures from oomycete species can be an arduous task, since many species do not produce spores [37] or need specific and labor-intensive requirements. For these cases, the mycelium represents a useful alternative for efficacy assessment. Different *Phytophthora* and *Pythium* spp. Were screened using a liquid medium inoculated with mycelium plugs, and the assessments of mycelial growth were performed both visually [27,29,31] and spectrophotometrically [32]. To overcome the use of mycelium plugs, Noel and collaborators [33] proposed a high-throughput method based on OD measurements for *Pythium* and *Phytophthora* species, using mycelial macerates as the source of inoculum; however, little information on inoculum standardization in the initial OD measurement has been reported. 

A further source of information regarding the high-throughput evaluation of active substances is represented by patent literature. Indeed, the Patent Cooperation Treaty set as a requirement a clear and complete description of the invention to ensure reproducibility from a person skilled in the art [38]. For this reason, several descriptions of high-throughput procedures are reported in the context of patenting; however, in practice, limited descriptions are often reported, thus not guaranteeing the same standard of reproducibility in patents as in scientific publications [39,40,41].

In contrast, for species belonging to the Fungi kingdom, high-throughput methodologies for antifungal susceptibility testing are widely reported in the literature, and different institutes worldwide have set standardized technical aspects at the national and international levels [42]. Despite the validity of these methodologies, protocols are mainly designed for fungal species with clinical relevance, and limited applications for filamentous fungi have been reported [42,43]. Moreover, oomycetes exhibit peculiar biological characteristics [44,45] (e.g., lack of septa, hyaline hyphae, motile spores, and different metabolic pathways) that significantly impact their requirements and must be considered in order to achieve proper manipulation and reproducible results. For this reason, setting up common standards for anti-oomycete susceptibility testing is desirable in the coming years.

The use of appropriate statistical analyses is crucial in scientific experiments, but very often, little importance has been given to this aspect when testing antifungal efficacy on oomycete species (Table 1). On the one hand, traditional parametric analyses such as one-way or two-way ANOVA can be used to analyze continuous responses such as OD values or colony diameters, but many assumptions are required to correctly employ these tests [46]. For instance, the normality assumption is crucial, and very often, researchers try different data transformations to improve normality [47,48]. This fact may cause trouble in the interpretation of inferred results, since parameter estimates cannot be easily interpreted in terms of the original response [49]. An alternative can be the use of nonparametric assays, which have fewer requirements in terms of assumptions and allow an easier interpretation of the results [48,50]. However, these statistical tests are less powerful with respect to their parametric counterparts and are very sensitive to outliers [51]. Very often, for both fungicide efficacy assays and dose–response analyses, it is convenient to express response data as proportions or percentages, which are easily interpretable, allowing researchers to make comparisons among different experiments and assay methodologies. These kinds of data are usually subjected to arcsine square root transformation prior to parametric analysis, causing misleading interpretations [52,53]. The use of beta regression can overcome these limitations, providing an appropriate method for modelling proportions [49]. Estimated marginal means (EMMs; sometimes called least-squares means) are values based on a model and represent the average response variable for each level of the predictor, thus allowing the performance of comparisons or contrasts considering both the main effects and the different factors/blocks belonging to the same model with a very interpretable output [54,55]. However, to date, the application of these techniques is still limited [56].

The aims of the present study were: (i) to provide a comprehensive comparison among two different multiwell-based efficacy screening methods for *P. infestans* using the CAA fungicide mandipropamid as the reference inhibitory molecule and (ii) to describe a discriminatory-dose assay focused on *P. ultimum* inhibition screening using mycelial suspensions and OD measurements, validated using a commercial fungicide, the phenylamide metalaxyl-M. The two types of efficacy approaches (i.e., dose–response analysis and discriminatory-dose assay) were adopted to cover the different possible needs of a researcher. The results of the proposed methods were compared to those of standard (reference) methods also employing appropriate statistical analyses (beta regression).

## 2. Materials and Methods

### 2.1. Fungal Material and Chemical Reagents

The *P. infestans* isolate n. 111344 (CBS; Baarn, The Netherlands) was grown on pea agar medium (PAM, 12.5% *w*/*v* frozen peas in distilled water and 1.2% *w*/*v* bacteriological agar). To obtain sporangia suspensions, 7-day-old *P. infestans* mycelium grown on Rye B agar plates [57] were flooded with ice-cold pea broth medium (PB, 12.5% *w*/*v* frozen peas in distilled water) and rubbed with a glass rod to liberate the sporangia. The suspension was then filtered through a 45 μm nylon mesh, counted in a Kova chamber (Kova Inc., Garden Grove, CA, USA), and adjusted to a final concentration of 2 × 10^4^ sporangia/mL.

*P. ultimum* isolate (n. 724.94; CBS; Baarn, The Netherlands) was cultured on PDA (Potato Dextrose Agar) plates (Liofilchem, Italy). Mycelial suspensions were prepared by homogenizing with a glass Potter tissue grinder (Thermo Fisher Scientific, Monza, Italy) in sterile PDB (Potato Dextrose Broth) amended with 0.1 g/L of Bacteriological agar (Difco, Franklin Lakes, NJ, USA), the mycelia from 4-day cultures, which had been grown on PDA plates overlaid with a sterile cellophane sheet. All the cultures were maintained at 20 °C in the dark.

Commercial fungicides Pergado SC (containing 250 g/L mandipropamid) and Ridomil Gold SL (465 g/L metalaxyl-M) (Syngenta, Milano, Italy) were employed in this study against *P. infestans* and *P. ultimum*, respectively. To assess *P. infestans* dose–response curves, five mandipropamid concentrations ranging from 0.1 to 1000 µg/L (logarithmic scale) were used. Metalaxyl-M was employed at 1000 mg/L to assess its activity against *P. ultimum,* using the dose suggested by the product’s label for field applications. Dilutions were prepared in the appropriate culture medium starting from 1000× stock solutions in water (400× for metalaxyl-M). Untreated controls with medium and sterile distilled water were also included in the assays.

### 2.2. Testing Procedures

The efficacy of the fungicides was tested on solid (9 cm Petri dish and 24-well plates with 1.5 cm diameter wells) and liquid media (96-well microtiter plates) (Figure 1). The Petri dish method represents the standard reference. Table 2 offers a resume of volumes and inoculum sources employed in each assay.

For Petri dish assays, three plates per fungicide concentration were prepared, inoculated with 5 mm mycelial plugs, incubated at 20 °C, and then used to measure the diameter of each colony. For *P. infestans* (Figure 1A)*,* each Petri dish was filled with 25 mL PAM medium amended with fungicide and inoculated with three mycelial plugs from the edges of actively growing cultures. Three diameters (mm) per colony were measured at 4 days post-inoculation (dpi). For *P. ultimum* (Figure 1B), which is a fast-growing species, a single mycelial plug was inoculated on each of the three Petri plates containing 25 mL of PDA medium, and three diameters (mm) per colony were measured at 4 days post-inoculation (dpi).

For 24-well plates assays, each well was filled with 1 mL of PAM (for *P. infestans*) or PDA (for *P. ultimum*) and inoculated with one mycelial plug. Four wells were used as replicates for each fungicide dose. 

Liquid culture assays were performed in 96-well flat-bottomed microtiter plates (Sero-well; Barloworld Ltd., Sandton, South Africa) by placing in each well 50 μL of *P. infestans* sporangia suspension and 50 μL of PB medium amended with fungicide by using a multi-channel pipette. Four wells were used as replicates for each fungicide dose. The absorbance (OD_620 nm_) of each well was measured (Sunrise Absorbance Reader; Tecan Group Ltd., Melbourne, Austria) at the start of the experiment and every 24 h for three consecutive days. Non-inoculated wells were also included in the assays as blank measurements. The same approach was employed for *P. ultimum* with the following modifications: PDB medium added with 0.1 g/L Bacteriological Agar (Difco, Franklin Lakes, NJ, USA) was used instead of PB, and mycelial suspensions were adjusted to OD_620_ = 0.050 before use to standardize the inoculum density; in this case, OD measurements were taken up to 4 dpi. All the assays described were carried out twice in two successive experiments. 

### 2.3. Growth Inhibition Calculation

Net mean diameters (without mycelial plug) for each colony were used as the response variable in solid media assays. Growth inhibition percentage for solid cultures (GIP_S_) was calculated according to Formula (1).
(1)$$GIP_{S}=100-\frac{D_{x}*100}{D_{C}}$$
where *D_x_* is the mean net diameter for the x-colony, and *D_C_* is the mean net diameter of colonies grown on control plates/wells.

Growth inhibition percentage for liquid cultures (*GIP_L_*) was calculated as described by Vercesi and collaborators [58]: (2)$$GIP_{L}=\frac{\Delta_{AC}-\Delta_{AF}}{\Delta_{AC}}*100$$
where Δ*_A_* is the difference between absorbance values recorded at t_0_ and t_3_ for control (Δ*_AC_*) and fungicide (Δ*_AF_*)-treated samples. To simplify the calculations, OD_620 nm_ values were multiplied by 1000. 

### 2.4. Statistical Analyses

Growth inhibition percentage (GIP) values can be assumed to be a sample from beta distribution; indeed, mycelial growth inhibition can be considered as a continuous proportion with respect to the untreated control mean response. GIP retrieved from mandipropamid-treated cultures were expressed as proportions (x) and transformed to include extreme values (0 and 1) as y = (x * (n − 1) + 0.5)/n, with n = sample size [59]. Then, a multiple beta-regression model (BRM) was fitted on proportion responses (Equation (3)) using betareg() [60]:(3)$$g\left(y\right)=\beta_{0}+\beta_{i,j}X_{i,j}$$
with *β* = intercept and *β*_0_ = the model’s intercept; *i* = (*i*th dose; 0.1, 1, 10, 100, and 1000 mg/L); *j* = (*j*th assay; 24 wells, 96 wells, and 9 cm Petri); g(•) = logit as the link function; and a constant φ (precision parameter). The X_ij_ parameter is represented by two categorical factors; for this reason, to compare among different combinations (dose–assay), estimated marginal means (EMMs) were retrieved from the fitted model over the *X_ij_* variables via emmeans(). The overall effect of independent variables was tested by the ANOVA-like type III test performed through a joint_test(). In addition, pairwise comparisons among estimated EMMs were calculated through the pairs() function in the emmeans package [55]. 

The dose–response analysis (DRA) on mandipropamid-treated cultures was assessed by fitting a three-parameters logistic regression using the drc package [61] on GIP values, grouping the independent variable according to assay type (24 wells, 96 wells, and 9 cm Petri). Model selection was performed according to the mselect() function output, using log-likelihood, Akaike Information Criteria (AIC), and residual sums of squares. Simultaneous inference of the estimated parameters was evaluated according to *t*-statistic *p*-values using the summary.drc() function, whereas, to assess the model fit, a lack of fit test was applied through drc::modelFit() [62]. Effective concentrations 50 (Relative EC_50_), representing the fungicide concentration able to inhibit colony growth by 50% compared to the untreated control, were estimated from each fitted model using ED.drc().

A multiple linear regression (MLR) model was employed to analyze the OD_620_ values (*Y*) registered in 96-well plates from the *P. ultimum* mycelial suspension inoculated wells during the experimental timeframe by using Equation (4), which can be reparametrized as Equation (5), resulting in exponential regression.
(4)$$log\left(Y\right)=\beta_{0}+\left(\beta_{1}X_{1}\right)+\left(\beta_{2}X_{2}\right)$$
(5)$$Y=e^{\beta 0}+e^{\beta_{1}X_{1}}+e^{\beta_{2}X_{2}}$$
where *β*_0_ represents the model’s intercept; *β*_1_*X*_1_ is the term related to the experimental timeframe, expressed as dpi; and *β*_2_*X*_2_ is the term related to the fungicide treatment (metalaxyl M treated or untreated). The MLR model was fitted through the lm() function in stats. To compare among different curves, EMMs at 2 dpi were retrieved from the fitted model over the *X*_2_ variable, and pairwise comparisons through the t.ratio test were calculated as previously described. To account for differences in OD_620_ values recorded at different dpi for the same treatment, a Kruskal–Wallis test followed by a Fisher’s least significant difference *post hoc* test using kruskal() in the agricolae package [63] was performed. All the statistical analyses were performed using R [64] in R studio 9.1 [65].

## 3. Results

### 3.1. P. infestans Response to Mandipropamid

The GIP and response values obtained from each test are reported in Table 3. Overall, with concentrations up to 1 µg/L, little or no reduction compared to the untreated control was observed, whereas 10 µg/L mandipropamid reduced *P. infestans* growth, with GIP values close to 80% in all three testing methodologies. Higher concentrations (100 and 1000 µg/L) completely inhibited the oomycete growth, with GIP values greater than 95% (96-wells assay) or equal to 100%. Notably, at 0.1 and 1 µg/L, mandipropamid determined an increase in diameter in the 9 cm Petri assay (hormetic effect). The growth curves obtained from absorbance measurements (OD_620_) defining the 96-wells assay showed a logarithmic trend at nonlethal concentrations (0–1 µg/L), whereas little or no growth was observed at higher concentrations (Figure 2A).

The BRM employed well-predicted GIP values over the different fungicide doses and assays (Pseudo R^2^: 0.95; RMSE: 0.044) (Appendix A), and for this reason, the EMMs turned out to be a good estimator for multiple comparisons. Overall, the effect of the assay was not significant according to the ANOVA-like test (χ^2^ = 2.79; df = 2; *p*-value = 0.06), whereas significant differences among doses were found (χ^2^ = 3167.3; df = 4; *p*-value < 0.001), and pairwise confrontations results are reported in Table 3, whereas estimated relative EC_50_ values are reported in Table 4.

### 3.2. P. ultimum Response to Metalaxyl-M

On solid media, metalaxyl-M totally inhibited *P. ultimum* mycelial growth after 3 dpi (Table 5). Total inhibition was also achieved against mycelial suspensions in the liquid medium; indeed, no visible growth was observed in treated wells after 3 dpi; moreover, absorbance values (OD_620_) measured up to 4 dpi for treated wells showed a flat trend compared to the untreated ones (Figure 3A). The MLR model (Figure 3B) well described the absorbance dynamics over time (adjusted R^2^ = 0.58), and the pairwise comparisons among EMMs retrieved at 2 dpi significantly differed (t.ratio = 1.32, df = 37, *p*-value < 0.0001), thus confirming the different behaviors in growth curves associated with treated and untreated wells. Moreover, no significant differences in OD_620_ values were recorded for treated wells from 1 to 4 dpi (Figure 3A), thus suggesting that mycelial growth was totally inhibited.

## 4. Discussion

In this work, two different kinds of assays were developed and tested, taking into consideration the pathogen features. A dose–response analysis was carried out for *P. infestans*, using sporangia as the source of inoculum, whereas a discriminatory dose assay was instead preferred for *P. ultimum*, performing a simple and fast test using mycelium macerates with proper inoculum standardization. The growth inhibition values obtained for *P. infestans* from the three different assays employed in this study exhibited comparable values, as also confirmed by the statistical analyses (BRM). Moreover, the dose–response curves described by the logistic regression resulted in a similar shape, with close relative EC_50_ values confirming the interchangeability between methods. The great advantage of replacing the 9 cm Petri dishes assay with the 24- and 96-wells assays relies on the possibility of reducing the amount of the antifungal compound up to 25 and 250 times (Table 2), respectively. This allows the researchers to achieve trustworthy results while reducing the amount of media and plastics needed for the assays, the costs for the assay, and fungicide waste disposal. The comprehensive resume of the efficacy percentages (expressed as proportions) obtained through the BRM and the use of EMMs for the multiple comparisons allowed us to compare simultaneously the two factors (assay type and fungicide concentration). This approach has several advantages compared to the common statistical analyses (e.g., ANOVAs and Kruskal–Wallis); indeed, it does not require any data transformation, handling proportions naturally, and it can address nonconstant dispersion for covariate values (useful in large experiments) through the precision parameter [66], and it can provide an easily interpretable output (Appendix A) through the use of EMMs. The latter represents a very versatile tool for multiple comparisons, since single or multiple factors can be considered starting from the same model (i.e., one-way and two-way data can be handled in the same model) and, since they are equally weighted according to the factor level, can compensate for the imbalance in the data, which is a very common issue in large biological experiments (i.e., presence of outliers) [55].

The use of metalaxyl-M successfully inhibited the *P. ultimum* mycelial growth both in solid and liquid cultures. In this case, the 24-wells assay can represent a valid alternative to the reference standard only for assays meant to establish the molecule’s efficacy with a binomial response (growth vs. no growth). Indeed, the mycelial growth in the control wells is strongly limited by their size and cannot provide useful information as a continuous response. The high-throughput method proposed by Noel and collaborators [33] used mycelial macerates as the source of inoculum, but different standardizations of mycelial seedlings (i.e., inoculum quantification) have been reported. Since the use of mycelial fragments in OD measurements is subject to higher standard deviations compared to the use of spores [67], the initial status of a mycelial suspension must be accounted to ensure reproducibility. Moreover, to avoid any problems of fragment decantation in the early phases, which can lead to absorbance reduction, a small amount of agarose was added to the medium [68,69]. Homogenized hyphal biomass has been proven to correlate linearly with dry cell weight [70]. The logarithmic growth behavior associated with *P. ultimum* mycelial suspensions in liquid cultures has been already described by Rawn and Van Etten [71]. For these reasons, the sigmoidal growth curve obtained from untreated wells through the OD_620_ monitoring confirmed successful growth monitoring using *P. ultimum* mycelial suspensions and also confirmed the possibility of determining the activity of antifungal substances and of dose–response analyses using validated approaches [33]. Nonlinear regression models (e.g., logistic, Gompertz, and Weibull) are often employed to model dose–response data [72], as is also done in the context of this study for mandipropamid-treated cultures. Microbial growth curves (i.e., growth over time) are often characterized by a sigmoidal trend that can be effectively predicted using the previously cited models. However, the reduced flexibility compared to linear models and the lack of an analytical solution for parameter estimations can badly reflect on the routine use. Indeed, a large library of functions is often necessary for model selection, and starting values are required to successfully fit the model, thus complicating the analyses [73]. In this context, the MLR employed in this study provided a simple and easy-to-interpret tool to statistically compare the *P. ultimum* growth curves based on OD data. For instance, the linear approximation reduced the complexity associated with the sigmoidal trend and allowed the use of EMMs for multiple comparisons (at midpoint time), with all the advantages previously described. 

Overall, the results obtained in this study suggest that both of the multiwell assays provided analogous information on antifungal compound efficacy in comparison with the standard reference method, thus reducing the amount of molecules needed and increasing the throughput. As pointed out by Hunter and collaborators [32], the use of colony diameter as a response variable does not allow the evaluation of the mycelial density and of the aerial growth habit that can occur in some species [74]. The use of dry weight can overcome these limitations, but it is also more time consuming and with limited throughput. In addition, the use of solid media can underestimate the effect of the antifungal compound due to the lower surface area of mycelium exposed [75] or because of the interaction between cationic antimicrobials and negatively charged agarose.

Considering the limitations associated with traditional techniques and the results highlighted in the present and in similar studies [32,33], the use of multiwell plates for anti-oomycete efficacy screenings should be encouraged. In this context, coupling the liquid medium and OD measurement allowed a more comprehensive estimation of mycelial growth while limiting the costs and speeding up the response assessment, setting a milestone for large experiments requiring growth monitoring such as those related to the fitness evaluation of oomycete structures belonging to different strains or species. These methods would also allow the assaying of the effect of antifungal compounds belonging to different classes and to compare the results obtained on different pathogen structures by the same compound more effectively. In addition, multiwell plates can be easily coupled with the use of robotic systems designed to automate repetitive tasks (e.g., pipetting), thus increasing the reproducibility and facilitating the use of full factorial designs [76]. Moreover, the pipetting operations performed on liquid medium are far easier and more precise compared to the methods needed for the solid one, facilitating the process automation. Finally, the use of spore or mycelial suspensions limit the need for specialized mycology trained personnel and equipment associated with the mycelium plug transfer. 

The present study aimed to set up a starting point for the establishment of common practices in anti-oomycetes compounds screening, with particular attention to young researchers who approach this field of study. To ensure the highest level of reproducibility, a careful description of the materials and methods, including laboratory practices and data analysis, have been provided for the reader. To endorse reproducibility, strains available in a fungal collection, as well as available commercial products containing known inhibitors, were employed in this study. However, the establishment of common practices is a long process that involves different expertise and needs. Therefore, limitations in the presented protocols must be addressed in the near future. Further evidence deriving from the use of biological replicates (i.e., different strains and species), preferably characterized by distinct phenotypes and sensitivity profiles, and the use of inhibitors with different modes of action should be tested in order to improve the reproducibility. Moreover, further validation of the use of mycelial fragments should be carried out through the dose–response analysis. Additionally, it must be pointed out that the high-throughput methodologies here described aim to investigate the inhibitory effect against cultivable pathogens on synthetic media. Usually, these kinds of tests are performed for the early-stage screening of new compounds, or they are applied in routine tests for fungicide resistance monitoring. However, pathogenesis is a complex phenomenon involving the interaction between the plant and the pathogen, and for this reason, assays involving the host’s susceptible tissues, either *ex vivo*, *in vivo*, or *in planta*, must be carried out for the complete validation of crop protection products.

## Figures and Tables

**Figure 1 microorganisms-11-00350-f001:**
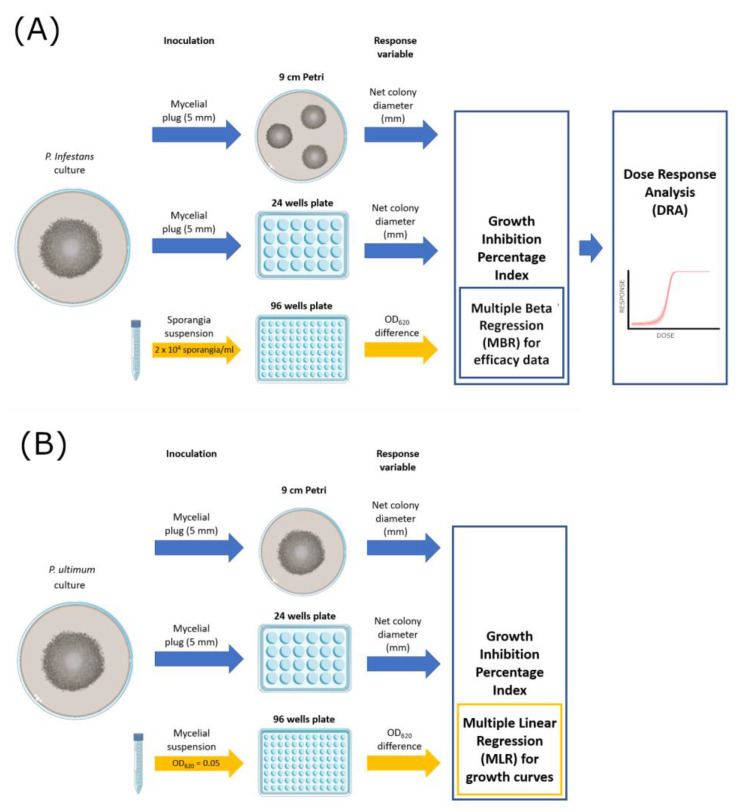
Graphical scheme of the experimental activities carried out on *P. infestans* (**A**) and *P. ultimum* (**B**).

**Figure 2 microorganisms-11-00350-f002:**
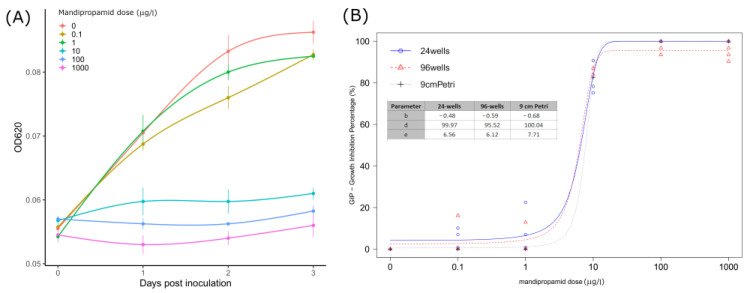
(**A**) Mean OD_620_ absorbance values measured in *P. infestans* sporangia inoculated wells from 0 to 3 dpi (24 h interval) with standard deviation (vertical bars). Lines were obtained through general additive model (GAM) regression fitting using the gam() function in R. (**B**) Curves obtained from the three-parameters logistic model predictions by analyzing GIP (growth inhibition percentage) indexes calculated for each assay (24 wells, 96 wells, and 9 cm Petri) at given mandipropamid concentrations. Points display GIP values recorded; estimated parameters for each curve are also reported.

**Figure 3 microorganisms-11-00350-f003:**
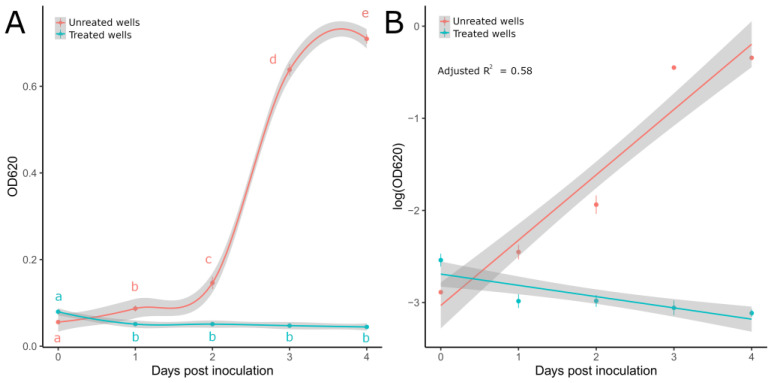
(**A**) OD_620_ values recorded from 0 to 4 dpi for *P. ultimum* mycelial suspensions treated and untreated with metalaxyl M in 96-well microtiter plates. Lines were obtained through general additive model (GAM) regression fitting using the gam() function in R. Letters display significant differences individuated by the Kruskal–Wallis test with LSD post hoc analysis for treated (KW = 11.6, df = 4, *p*-value = 0.02) and untreated wells (KW = 18.29, df = 4, *p*-value = 0.001). (**B**) Regression lines obtained from the MLR model on log-transformed OD values. Light-colored ribbons displaying 95% confidence intervals are reported. Dots show mean values with standard deviations (vertical lines).

**Table 1 microorganisms-11-00350-t001:** Resume of high-throughput methodologies employed to assay the efficacy of fungicides against different oomycetes species.

Reference	Species	Method	Source of the Inoculum	Response Variable	Comparison with Reference Method	Validation on Known Inhibitors	Statistical Analyses on Efficacy Data
[24]	*Phytophthora nicotianae*	96-well plates with liquid medium	Zoospores	Optical Density (OD620 nm)	Yes	Yes	Two-way ANOVA on log-transformed growth measurements; logistic model to determine EC_50_ concentrations
[25]	*P. infestans*	96-well plates with liquid medium	Zoospores	Optical Density (OD630 nm)	No	No	Not employed
[26]	*Phytophthora sojae*	Lab-on-a-Chip (LOC) Platform	Zoospores	Germination and germling growth	No	Yes	Not employed
[27]	*Phytophthora* spp.	48-well plates with solid medium	Mycelial plugs	Visual assessment (0–5 scale)	Yes	Yes	Not employed
[28]	*P. infestans* and *Phytophthora capsici*	96-well plates with liquid medium	Sporangia and zoospores	Optical Density (OD630 nm) and red fluorescence (excitation at 360 nm, emission at 465 nm)	No	No	Not described
[29]	*Pythium* spp.	48-well plates with solid medium	Mycelial plugs	Visual assessment (0–5 scale)	No	Yes	Not employed
[30]	*Phytophthora agathidicida* and *Phytophthora cinnamomi*	Disk diffusion assay on solid medium	Mycelial plugs	Colony diameter	No	No	Nonlinear regression to determine EC_50_ concentrations
[31]	*Pythium irregulare*	24-well plates with liquid medium	Mycelial plugs	Visual assessment (0–5 scale)	No	Yes	Nonparametric statistics on rank transformed ordinal scale
[32]	*Phytophthora* spp.	24-well plates with liquid medium	Mycelial plugs	Optical Density (OD620 nm)	Yes	Yes	Generalized linear model (GLM) with logit link on proportions of inhibition; EC_50_ values estimated by visual interpretation of compound concentration (x-axis) and growth inhibition (y-axis)
[33]	*Pythium, Phytophthora*, and *Phytopythium* spp.	96-well plates with liquid medium	Mycelial fragments	Optical Density (OD600 nm)	Yes	Yes	EC_50_ values estimated as described by Ritz et al. 2015
[34]	*P. infestans*	96-well plates with liquid medium	Zoospores	Optical Density (OD610 nm)	No	No	Not employed

**Table 2 microorganisms-11-00350-t002:** Total volume of medium and inoculum sources employed for each kind of assay. Volume reduction compared to the reference method is also displayed.

Assay	Total Volume per Replica (mL)	Source of the Inoculum	Volume Reduction
*Solid medium (Petri dish)*	25	Mycelial plugs	-
*Solid medium (24 wells)*	1	Mycelial plugs	25×
*Liquid medium (96 wells)*	0.1	Sporangia (*P. infestans*) or mycelial (*P. ultimum*) suspensions	250×

**Table 3 microorganisms-11-00350-t003:** Response values (colony growth measured in mm or Δ_A_) and GIP indexes obtained for each assay (9 cm Petri, 24 wells, and 96 wells) at each mandipropamid concentration. Mean values with standard deviations for the two experiments are reported. Means sharing a letter are not significantly different (Sidak-adjusted comparison among Estimated Marginal Means—EMMs from the beta-regression model; alpha = 0.05).

MandipropamidConcentration (µg/L)	9 cm Petri		24 Wells		96 Wells	
	Response (mm)	GIP_S_	Response (mm)	GIP_S_	Response (Δ_A_)	GIP_L_
**0**	11.46 ± 0.45		10.75 ± 0.16		31 ± 3.67	
**0.1**	14 ± 1.87	0 ± 0 a	10 ± 0.47	6.98 ± 4.38 a	27 ± 1.41	8.06 ± 9.31 a
**1**	13.22 ± 0.65	0 ± 0 a	9.91 ± 1.1	7.75 ± 10.24 a	28.25 ± 4.25	3.22 ± 6.45 a
**10**	2 ± 0	82.56 ± 0.5 b	1.75 ± 0.87	83.72 ± 8.15 b	4.25 ± 0.5	86.29 ± 1.61 b
**100**	0 ± 0	100 ± 0 c	0 ± 0	100 ± 0 c	1.25 ± 0.95	95.97 ± 3.08 c
**1000**	0 ± 0	100 ± 0 c	0 ± 0	100 ± 0 c	1.5 ± 1.29	95.16 ± 4.16 c

**Table 4 microorganisms-11-00350-t004:** Mean relative EC_50_ values, standard errors (SE), and 95% confidence intervals for each assay (9 cm Petri, 24 wells, and 96 wells) estimated from the three-parameters logistic model. All the values are expressed as µg/L of mandipropamid.

Assay	Mean	SE	CI 95%
**9 cm Petri**	7.71	0.4	6.91–8.5
**24 wells**	6.56	0.32	5.92–7.19
**96 wells**	6.12	0.42	5.28–6.97

**Table 5 microorganisms-11-00350-t005:** Response values (expressed as mm or Δ_A_) and GIP indexes obtained for each assay type (24 wells, 96 wells, and 9 cm Petri) for treated (metalaxyl M—1000 mg/L) and untreated plates/wells. Mean values and standard deviations from the two experiments are reported.

Metalaxyl M Dose (µg/L)	9 cm Petri		24 Wells		96 Wells	
	Response (mm)	GIP_S_	Response (mm)	GIP_S_	Response (Δ_A_)	GIP_L_
**0**	10 ± 0.0		20.28 ± 0.47		582.50 ± 24.2	
**1000**	0 ± 0	100 ± 0	0 ± 0	100 ± 0	−23.25 ± 16.09	100 ± 0

## Data Availability

The R code used to perform the statistical analyses showed in the present study is available at the following link: https://github.com/demar2/Marciano2023_Microorganisms (accessed on 30 January 2023).

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
