# Peer review of "Methods for Fungicide Efficacy Screenings: Multiwell Testing Procedures for the Oomycetes Phytophthora infestans and Pythium ultimum"

_microorganisms, 2023, doi:10.3390/microorganisms11020350_

Round 1

Reviewer 1 Report

Thank you for a very interesting paper. Certainly of interest to the wider community. 

Page 1 Line 36 .. fungicide market fraction for oomycetes is publicly available. (There are existing reports on the crop protection market, but they are accessible against a fee) 

Kriskal-Wallis is written in many different ways. Line 243 - 244- 338

There is no reference to patent literature. Agrochemical companies must deliver the testing of oomycetes species in a high-throughput system, and they have an obligation to describe their invention, which means that some information should be available. Maybe worth adding a note from completeness. 

Author Response

Reviewer #1 (R1)

Authors (AU) Dear Reviewer, we wish to thank you for your interest in our study and the precious and punctual observations that arose from the careful reading of the manuscript. We made the following modifications in the hope that these clarifications could help the reader in a better understanding of the work done:

R1. “Page 1 Line 36. fungicide market fraction for oomycetes is publicly available. (There are existing reports on the crop protection market, but they are accessible against a fee)” 

Since several active compounds are both registered for oomycete and fungi control (e.g. QoIs and multisite products), it’s hard to separate oomycete market fraction. However, thank you for the suggestion about reports, we’ll investigate them carefully in the future. In any case, we reformulated the statement to avoid misinterpretations (line 36).

“Moreover, the market value of active ingredients available for oomycete control accounted for hundreds of millions of dollars in the USA market only for 2011, thus suggesting the relevant economic impact associated with these organisms [7].”

R1: “Kriskal-Wallis is written in many different ways. Line 243 - 244- 338”

AU. We uniformed the Kruskal-Wallis notation throughout the paper. As for line 244, “kruskal()” is referred to the specific function employed in R programming language, for this reason, we left it unchanged.

R1. “There is no reference to patent literature. Agrochemical companies must deliver the testing of oomycetes species in a high-throughput system, and they have an obligation to describe their invention, which means that some information should be available. Maybe worth adding a note from completeness.” 

As suggested, we added a small paragraph starting from line 96.

“A further source of information regarding the high-throughput evaluation of active substances is represented by patent literature. Indeed, the Patent Cooperation Treaty set as a requirement a clear and complete description of the invention, to ensure reproducibility from a person skilled in the art [38]. For this reason, several descriptions of high-throughput procedures are reported in the context of patenting; however, in practice, limited descriptions are often reported, thus not guaranteeing the same standard of reproducibility in patents as in scientific publications [39, 40, 41].”

Reviewer 2 Report

In this manuscript the authors compare three methods for determining the effect of fungicides on growth of the oomycetes Phytophthora infestans and Pythium ultimum. The goal is to identify a high throughput method that will use less media and less of the growth inhibitor than the standard method of amending agar medium in a 9 cm diameter petri plate. A 24-well plate and 96 well plate was used with P. infestans sporangia and P. ultimum homogenized mycelium. They concluded that each method resulted in similar results. Although the premise of the work is excellent, I believe publication is premature. As presented, the manuscript does not add significantly to the literature.

For the P. infestans experiment, the fungicide mandipropamid was diluted in 10-fold increments (0.1, 1, 10, 100, 1000 ug/L). The EC50 value was determined between 1 ug/L and 10 ug/L at 0% growth inhibition and 83% growth inhibition. This is inappropriate for determining EC50 values. Additional dilutions should have been tested between 1 and 10 ug/L. Apparently the experiment was not repeated and the results are from a single experiment (although means were determined from n=12 measurements for the petri plate assay and n=4 for the 24 well and 96 well plate assay. Additionally, the test should be done with at least one other fungicide to demonstrate that the multi-well plate assays can be applied broadly.

For the P. ultimum experiment, only one concentration of metalaxyl-M was used, which completely inhibited growth. A range of concentrations should be used to show that there is a linear response with homogenized mycelium. Again, it appears that the experiment was only done once with a single fungicide.

Other issues:

The English grammar needs to be polished throughout the manuscript. Photos in Figure 3 are of poor quality and do not add information to the paper. Similar assays have been validated for fungi and fungicides, but these were not discussed.

Author Response

Reviewer #2 (R2)

  1. Dear Reviewer, we wish to thank you for your interest in our study and the precious and punctual observations that arose from the careful reading of the manuscript. We made the following modifications in the hope that these clarifications could help the reader in a better understanding of the work done:

R2. “For the P. infestans experiment, the fungicide mandipropamid was diluted in 10-fold increments (0.1, 1, 10, 100, 1000 ug/L). The EC50 value was determined between 1 ug/L and 10 ug/L at 0% growth inhibition and 83% growth inhibition. This is inappropriate for determining EC50 values. Additional dilutions should have been tested between 1 and 10 ug/L.”

EC50 estimation depends on the fungicide concentration spectrum and on the corresponding biological response distribution. According to the literature, two definitions of EC50 are commonly accepted, absolute and relative EC50. The relative EC50 is defined as the concentration corresponding to a response midway between the estimates of the lower and upper asymptotes retrieved from the fitted model. For logistic-based models, the midpoint corresponds to the “e” parameter, therefore, the accurate estimation is only dependent on “b” and “d” values (corresponding to the asymptote values on the y-axis). Moreover, as common practice standard, we can assume an accurate estimation of asymptote when the responses from at least two concentrations are in the asymptotic trend. For this reason, often five concentrations of fungicide, plus the untreated control are used as we did in the present study.

The criticism that arose from the reviewer is formally correct, and in the case of absolute EC50 estimation, the suggestion of using additional concentrations must be addressed for a proper determination. However, the purpose of the assay wasn’t the punctual determination of the absolute EC50 but rather the evaluation of the overall dose-response relationships obtained from the different methodologies. To summarize the results in an effective way we chose relative EC50 and related statistics (Std. error and confidence intervals). To overcome this issue, we modified the EC50 notation throughout the text, referring to it as “relative EC50”.

R2. “Apparently the experiment was not repeated, and the results are from a single experiment (although means were determined from n=12 measurements for the petri plate assay and n=4 for the 24 well and 96 well plate assay.”

The number of measurements is correct. However, we guilty missed specifying that the means reported are obtained from two different and subsequent experiments. We added a sentence in line 200 for further clarification.

“All the assays described were carried out twice, in two successive experiments.”

R2. “Additionally, the test should be done with at least one other fungicide to demonstrate that the multi-well plate assays can be applied broadly.”

We assessed the issue by adding a statement starting from line 432.

“Therefore, limitations in the presented protocols must be addressed in the near future. Further evidence deriving from the use of biological replicates (i.e. different strains and species), preferably characterized by distinct phenotypes and sensitivity profiles, and the use of inhibitors with different modes of action should be tested in order to improve the reproducibility”.

R2. “For the P. ultimum experiment, only one concentration of metalaxyl-M was used, which completely inhibited growth. A range of concentrations should be used to show that there is a linear response with homogenized mycelium.”

In this work, we specifically aimed at developing two different assays: a dose-response assay and a discriminatory dose assay. We specified this in the introduction section by writing “The aims of the present study were: (i) to provide a comprehensive comparison among two different multiwell-based efficacy screening methods for P. infestans, using the CAA fungicide mandipropamid as reference inhibitory molecule; (ii) to describe discriminatory-dose assay focused on P. ultimum inhibition screening using mycelial suspensions and OD measurements, validated using a commercial fungicide, the phenylamide metalaxyl-M.”

We tested the discriminatory dose analysis on P. ultimum because it’s a more complicated case in the sense of inoculum standardization (we work with mycelium and not with spores), and the need for a simple and effective test is higher. We clarified this point at the beginning of the discussion by writing:

“In this work, two different kinds of assays were developed and tested, taking into consideration the pathogen features. A dose-response analysis was carried out for P. infestans, using sporangia as the source of inoculum, whereas a discriminatory dose assay was instead preferred for P. ultimum, performing a simple and fast test using mycelium macerates with proper inoculum standardization.”

Since we chose a discriminatory dose assay, we only employed a single concentration according to fungicide label (at this concentration the fungicide should completely inhibit the fungal growth). We agree with the comment on the need of testing more concentrations and develop a dose response analysis. Indeed, homogenized hyphal biomass has been proved to correlate linearly with dry cell weight and the logarithmic growth behavior associated with P. ultimum mycelial suspensions in liquid cultures, has been already described by Rawn and Van Etten [63]. We added a statement from line 437 to clarify this deficiency.

“Moreover, further validation of the use of mycelial fragments should be carried out through dose-response analysis.”

R2. “Photos in Figure 3 are of poor quality and do not add information to the paper”

We removed Figure 3 as suggested.

R2. “Similar assays have been validated for fungi and fungicides, but these were not discussed”

To clarify this aspect, we added a small paragraph starting from line 103

“In contrast, for species belonging to the Fungi kingdom, high-throughput methodologies for antifungal susceptibility testing are widely reported in the literature, and different institutes worldwide have set standardized technical aspects at national and international levels [42]. Despite the validity of these methodologies, protocols are mainly designed for fungal species with clinical relevance, and limited applications for filamentous fungi have been reported [42; 43]. Moreover, oomycetes exhibit peculiar biological characteristics [44, 45] (e.g., lack of septa, hyaline hyphae, motile spores, and different metabolic pathways) that significantly impact their requirements and must be considered in order to achieve proper manipulation and reproducible results. For this reason, setting up common standards for anti-oomycete susceptibility testing is desirable in the coming years.”

Reviewer 3 Report

This study was focused on the testing and validation of screeening protocols for determining the in vitro anti-oomycete fungicide efficacy using the plant pathogens Phytopathora infestans and Pythium ultimum as models. The study was based on dose-response (to determine the EC50s) or discriminatory doses analyses. The introduction brought a thorough literature review of the state-of-the-art high throughput methologies to assay the in vitro fungicide efficacy for oomycetes. It is very well written and brings a very smooth flow of thoughts, leading to a very clear rationale for the research and the hypothesis testing.

The materials and methods section is very detailed and descriptive of the methodologies employed, including very proper illustrations.

The results also were well presented and summarized, with figures, tables and statistiscal analysed supporting the study´s overall observations.

The authors concluded suggesting that both the multiwell assays tested provided analogous information on antifungal compounds efficacy in comparison with the reference method, allowing for reducing the amount of the chemical fungicide molecule needed and increasing the throughput of the assay.

Although they approached the limitations of their methodology in the discussion, not all limitations were included. It would have been very important to emphazise also a limitations that was not addressed, which was the limited sample size: only one isolate representing each one of the two oomycete species used as models. For studies such as this, a biological replicate (a triplicate would be even better) is warranted to show reproducibility. Besides, strains / isolates with contrasting sensitivity phenotyping should have been included. Please address that. 

Finaly, it is also important to include information on how reproducible is the high-throughput fungicide sensitivity in vitro assays with an in vivo (in planta) assay.

For this reason I am recommending acceptance only after this minor issue has been addressed.

Author Response

Reviewer #3 (R3)

R3. “Although they approached the limitations of their methodology in the discussion, not all limitations were included. It would have been very important to emphazise also a limitations that was not addressed, which was the limited sample size: only one isolate representing each one of the two oomycete species used as models. For studies such as this, a biological replicate (a triplicate would be even better) is warranted to show reproducibility. Besides, strains / isolates with contrasting sensitivity phenotyping should have been included. Please address that.”

We assessed the issues by adding a paragraph starting from line 422.

“The present study aimed to set up a starting point for the establishment of common practices in anti-oomycetes compounds screening, with particular attention to young researchers who approach this field of study. To ensure the highest level of reproducibility, a careful description of materials and methods, including laboratory practices and data analysis, have been provided for the reader. To endorse reproducibility, strains available in a fungal collection as well as available commercial products containing known inhibitors were employed in this study. However, the establishment of common practices is a long process that involves different expertise and needs. Therefore, limitations in the presented protocols must be addressed in the near future. Further evidence deriving from the use of biological replicates (i.e. different strains and species), preferably characterized by distinct phenotypes and sensitivity profiles, and the use of inhibitors with different modes of action should be tested in order to improve the reproducibility”

R3. “Finaly, it is also important to include information on how reproducible is the high-throughput fungicide sensitivity in vitro assays with an in vivo (in planta) assay.”

A clarification on this point has been added from line 440

“Usually, these kinds of tests are performed for the early-stage screening of new compounds, or they are applied in routine tests for fungicide resistance monitoring. However, pathogenesis is a complex phenomenon involving the interaction between the plant and the pathogen, and for this reason, assays involving the host’s susceptible tissues, either ex vivo, in vivo or in planta, must be carried out for complete validation of crop protection products.”

Round 2

Reviewer 2 Report

The authors have addressed my scientific concerns from the previous review. English usage is improved, although grammatical errors and awkward phrasing are still present.  

Author Response

Dear Reviewer, we wish to thank you for your appreciation of the improved version of the manuscript. As we already mentioned with the academic editor the text was proofread by a British native-speaker expert. However, if you notify us of grammar errors or awkward phrasing we can try to solve them accordingly.

Attached to this message you can find the declaration made by the external expert.

Best regards,
Demetrio Marcianò and Silvia Laura Toffolatti
